# An Improved, Assay Platform Agnostic, Absolute Single Sample Breast Cancer Subtype Classifier

**DOI:** 10.3390/cancers12123506

**Published:** 2020-11-25

**Authors:** Mi-kyoung Seo, Soonmyung Paik, Sangwoo Kim

**Affiliations:** 1Department of Biomedical Systems Informatics, Brain Korea 21 PLUS Project for Medical Science, Yonsei University College of Medicine, Seoul 03722, Korea; mkseo82@yuhs.ac; 2Severance Biomedical Science Institute, Yonsei University College of Medicine, Seoul 03722, Korea

**Keywords:** breast cancer, subtyping, classifier, machine learning, optimization

## Abstract

**Simple Summary:**

A platform and normalization bias agnostic single sample classifier (SSC) is needed. A random forest SSC (MiniABS) utilizing 11 genes was developed and validated. MiniABS surpassed previous classifiers and can be applied on multiple platforms. MiniABS may provide a cost-effective strategy for the diagnosis of breast cancer.

**Abstract:**

While intrinsic molecular subtypes provide important biological classification of breast cancer, the subtype assignment of individuals is influenced by assay technology and study cohort composition. We sought to develop a platform-independent absolute single-sample subtype classifier based on a minimal number of genes. Pairwise ratios for subtype-specific differentially expressed genes from un-normalized expression data from 432 breast cancer (BC) samples of The Cancer Genome Atlas (TCGA) were used as inputs for machine learning. The subtype classifier with the fewest number of genes and maximal classification power was selected during cross-validation. The final model was evaluated on 5816 samples from 10 independent studies profiled with four different assay platforms. Upon cross-validation within the TCGA cohort, a random forest classifier (MiniABS) with 11 genes achieved the best accuracy of 88.2%. Applying MiniABS to five validation sets of RNA-seq and microarray data showed an average accuracy of 85.15% (vs. 77.72% for Absolute Intrinsic Molecular Subtype (AIMS)). Only MiniABS could be applied to five low-throughput datasets, showing an average accuracy of 87.93%. The MiniABS can absolutely subtype BC using the raw expression levels of only 11 genes, regardless of assay platform, with higher accuracy than existing methods.

## 1. Introduction

During the past two decades, a series of meta-analyses of prospective randomized clinical trials for early breast cancer conducted by the Early Breast Cancer Trialists’ Collaborative Group have produced definitive treatment guidelines [1]. In parallel, the description of intrinsic molecular subtypes by Perou et al. in 2000 revolutionized our understanding of the heterogeneity of breast cancer biology and its impact on natural history and treatment response [2].

Perou’s initial description of the intrinsic subtype was through an unsupervised clustering and did not provide a classifier for a new single sample (i.e., single sample classifier; SSC) outside the study cohort [2]. The intrinsic subtype classifier has evolved over time into its final form—PAM50 SSC [3]. PAM50 SSC assigns a case based on the nearest distance (closest correlation) to subtype centroids of the 50 most robust classifier genes constructed from a fixed reference patient cohort. Although PAM50 SSC has been adopted as the gold standard in The Cancer Genome Atlas (TCGA) and many other studies, PAM50 SSC has its limitations due to its relativistic nature. (1) Due to the need for normalization and gene-centering (standardization) of the study cohort data before measuring each sample’s distance to the centroids, PAM50 SSC assignment is influenced by the technology platform and the normalization method. For example, in the TCGA dataset, overall agreement for the PAM50 subtype determined by microarray versus RNA-seq was 83% [4]. (2) More importantly, the composition of the study cohort, especially the proportion of ER+ tumors (for example, a trial cohort of only HER2-positive patients) influences subtype assignments [5]. Therefore, PAM50 SSC is not an absolute classifier that can be applied to standalone single patient data.

While subtype-specific clinical trials have gained momentum along with the development of targeted therapies, eligibility for clinical trials is usually determined by clinical surrogate markers (such as ER immunohistochemistry), which only moderately correlate with intrinsic subtypes defined by gene expression profiling [6,7]. The correlative science aim of trials often includes determination of molecular subtypes but they are conducted employing diverse technology platforms including RNA-seq [4,8,9]. Eventually, trial results will often have to be subjected to meta-analyses to develop a new standard of care and cross compatibility of subtype determination will be important.

In parallel, in clinical practice, the decision to use chemotherapy for ER+ tumors is aided by gene expression-based prognostic or predictive assays such as OncotypeDx [10,11,12], Mammaprint [13], or Prosigna [9,14] tests, which were developed through supervised training with survival data. In meta-analyses with microarray data from a large cohort, all prognostic algorithms assigned the same ER+ tumors with low proliferation (i.e., Luminal A subtype) to a low risk class. Thus, the clinical utility of the prognostic gene expression-based tests originates from their ability to differentiate between the Luminal A versus Luminal B subtype among ER+ breast cancer, and therefore, risk assignment by these tests should be the same for each patient tested. However, in reality, agreement of risk assignments among these tests is less than 50% [15,16]. This surprisingly low agreement among prognostic tests stems from the fact that each clinical test uses a different gene expression measurement platform and more importantly, uses a proprietary within-sample data normalization technique, in order to be able to work with gene expression data from a single patient in contrast to a uniform, cohort-based normalization method used for meta-analysis.

Absolute Intrinsic Molecular Subtyping (AIMS), developed by Paquet et al. [4], was the first true SSC, since it only considers the absolute gene expression values of a given sample without referring to their relative expression levels within a cohort. AIMS uses 100 binary rules comparing absolute gene expression values of 151 genes to assign subtypes. While the agreement with PAM50 was ~77%, AIMS led to a more stable subtyping compared to PAM50 that was influenced by the composition of cohorts [4]. Despite its strength and technology platform independence, AIMS cannot be readily translated into clinical practice due to the requirement of measuring gene expression levels of 151 genes, which typically requires microarray, RNA-seq, or NanoString. These data underscore the clinical need to develop an absolute SSC that requires only a small number of genes and is technology platform-independent.

We developed MiniABS (Mini Absolute Breast Cancer Subtyper) using a Random Forest model of pairwise gene expression ratios among 11 functional genes. With a systematic gene selection and reduction step, we aimed to minimize the size of the gene set without losing the functional interpretability of the classifier. We validated the model’s performance using a large, heterogeneous cohort that consists of multiple public datasets across four different technology platforms. We anticipate that the high accuracy and reproducibility of MiniABS may provide an SSC at a low cost, as well as providing a method for cross comparison among gene expression datasets generated with different technology platforms for meta-analyses of clinical trials data.

## 2. Results

### 2.1. Overview of MiniABS

The main goal of MiniABS is to construct an absolute single sample classifier to subtype breast cancer with a minimal gene set. The entire process through which we developed MiniABS is illustrated in Figure 1. Machine learning classifiers were trained with a large-scale public database of breast cancer gene expression data annotated with PAM50 subtypes. To select informative genes for the classification of subtypes, a ranked list of genes significantly up- or down-regulated in only one of the five subtypes (subtype-specific differentially expressed genes, ssDEGs) was extracted by performing a Wilcoxon rank-sum test. We started from five 1st level genes from the ranked list of ssDEGs and stepwisely added the five genes from the next level down to the fifth level. Then, four well-known subtype marker genes, ESR1, PGR, ERBB2, and MKI67 [17,18], were used as seed genes and stepwisely, five genes were added from each level of the ssDEGs from the top five levels. For each input gene set, expression ratios between all possible combinations of gene pairs were log_2_(RPKM+1)-transformed and pairwise gene expression ratios (PGER) were calculated and used as actual inputs in the machine learning training (Appendix A). In the training phase, the classifier with the smallest gene set, but maximal classification power, was selected as an initial model, which was further tested on an independent validation cohort to confirm its accuracy. Detailed dataset information and acquisition processes are provided in Appendix A. When genes were ranked based on the importance of PGER in the initial model in the training phase, we noted a remarkable decrease in importance between the top 7 and 8 genes. Thus, model optimization to reduce the size of the gene set was performed by sequentially deducting genes one by one from the lowest rank (13th) to the 7th gene, after which the model was rebuilt without the removed genes in the same way applied in the training phase using the PGER matrices of these genes. The best performing model was selected as the final model for MiniABS. The TCGA test set was then used to test the accuracy of the finalized MiniABS, before moving on to the next step of independent validation across different technology platforms. We further tested it on an independent validation cohort to confirm its accuracy.

### 2.2. Training and Optimization of the Classifiers

The top-ranked ssDEGs from the TCGA BRCA dataset are shown in Table 1 (Appendix A for the full list). We found that over- or under-expression of MLPH (*q* = 8.2 × 10^−37^), FGFR4 (*q* = 4.3 × 10^−15^), CEP55 (*q* = 1.2 × 10^−46^), and KRT17 (*q* = 6.8 × 10^−17^) were identified as the top subtype-specific genes in Basal-like, Her2E, LumA, and LumB, respectively, with high statistical significances (Table 1). Similarly, subtype-specific expression of the four seed genes (ESR1, ERBB2, PGR, and MKI67) was confirmed (*q* = 1.96 × 10^−18^~7.95 × 10^−41^) (Appendix A).

By combining the ssDEGs and seed genes, five input gene sets were constructed (Figure 2A). Briefly, each set from a kth-level ssDEG subset consists of 4 + 5 k genes (seed genes + top-kth ranked ssDEGs from each of the five subtypes); however, the actual number was fewer due to overlap between seed genes and ssDEGs. For each input gene set, paired gene expression ratios (PGER) were calculated and fed into 28 classifiers, which encompassed four different machine learning algorithms (support vector machine (SVM) with radial kernel, classification and regression tree (CART), random forest (RF), and naïve Bayes (NB)) times seven distinct slack margin variables (α) that control the level of confidence.

The ssDEGs were identified by a Wilcoxon rank-sum test and ranked according to False Discovery Rates (FDR) per subtype. Colors denote up- (red) or down- (blue) regulated genes in comparison to median expression of the gene between the given subtype and the remaining subtypes.

Cross-validation of the classifiers showed robust accuracy of >80% for most of the trials (Figure 2B). Of the four machine learning algorithms, RF showed the best average performance (Appendix A). The effect of α was minimal, indicating a minimal dependency on experimental noise, thereby confirming the robustness of the classifier (Appendix A). The best accuracy of the first-level input gene set (*N* = 8) was 83.45%, which was further increased to 88.04% at the second level (*N* = 13), with an α of 1.00. The increase in the number of genes did not confer better accuracy after second-level input (indicated with a red arrow in Figure 2B). Therefore, we concluded that the most efficient classifier would not require more than 13 genes.

Next, we attempted to further reduce the number of genes required for classification. We measured mean decreases in Gini index values (the average decrease in node purity when a gene is removed) for the 13 genes and identified two distinctive groups: a highly informative group of seven genes and a less informative group of six genes (Figure 2C, 40.19–60.95 vs. 20.25–24.0). With the seven genes retained, we tested if removal of one of the less informative genes would lead to a drop in accuracy by building six more classifiers using 8 to 13 genes. We found that accuracy was maintained without two genes (88.26% and 88.24% without GRB7 and GRB7/KRT14, respectively, compared to 88.04% for 13 genes) and started to decrease after the removal of a third gene (Figure 2D, red arrow). Therefore, MiniABS was finally defined as an RF model of 11 genes (ESR1, PGR, ERBB2, MKI67, MLPH, FGFR1, CEP55, KRT17, FOXA1, MYBL2, and SFRP1).

Upon validation with the TCGA test set (20% of the discovery cohort, *N* = 85), MiniABS showed an accuracy of 88.24%, a 14.12% increase above AIMS (Figure 2E).

### 2.3. Biological Relevance of the PGERs

The actual feature components of MiniABS are PGERs, not a single gene expression level. We analyzed feature importance and the biological relevance of the 55 PGERs generated from 11 genes. The distribution of the top PGER values across the subtypes confirmed that the expression ratios were subtype-specific (Appendix A, P ranging 10^−58^–10^−59^).

The effects of PGERs on subtype specification were more clearly visible in two-dimensional (gene × gene) space (Appendix A), depicting subtype-specific clustering. We noted that Basal-like samples were well clustered in many gene pairs characterized by lower expression of FOXA1 and MLPH. This poses the possibility of building a much simpler classifier that only separates Basal-like from other subtypes. However, unlike Basal-like, there was no single PGER that could clearly separate the other four subtypes, implying that accurate discrimination cannot be achieved by a simple set of rules and requires a probabilistic model like MiniABS.

### 2.4. Validation on Independent Datasets

The most common pitfall in classification is model over-fitting, wherein the classifier is locally optimized to a training set and shows reduced reproducibility in independent datasets. To address this concern, we repeated our tests on an expanded validation set of 5816 samples from 10 independent studies with known PAM50 subtypes. As these studies were highly heterogeneous in their sizes, cohort compositions, data scales, and generation methods, consistently accurate classification of these sets would validate the robust performance of MiniABS (Appendix A and Appendix A).

First, we validated the performance of MiniABS in the largest single cohort dataset available for RNA-seq with clinical follow-up and treatment data (GSE96058, *N* = 3409) (Appendix A). There was a good agreement between MiniABS and the author-provided PAM50 subtypes (accuracy = 76.7%, kappa = 0.613; 95% CI = 0.590–0.635). Of note, MiniABS classified 70.0% of tumors as LumA compared to 50.1% by PAM50, and assigned only 0.1% to the Normal-like subtype. This was due to misclassification of 404/767 tumors from LumB and 208/225 Normal-like as LumA subtype. One obvious concern is the misclassification of LumB to LumA. However, there was no difference in survival of MiniABS LumA subtype patients versus PAM50 LumA subtype patients treated by endocrine therapy, even though more patients were classified as LumA by MiniABS (Figure 3A).

Intriguingly, endocrine therapy-treated patients who were classified as LumA by both PAM50 and MiniABS (*N* = 1082) had the best clinical outcome, and LumB by PAM50 alone (*N* = 139) had the worst outcome, whereas those with LumB by PAM50 but LumA by MiniABS (*N* = 209) showed intermediate outcomes (Figure 3B). Among 346 patients with LumB tumors by PAM50, the Cox model showed the trend for better outcome for those misclassified as LumA by MiniABS (*N* = 209) compared to those classified as LumB (*N* = 139) (HR = 1.5: 95% CI = 0.9–2.4, *P* = 0.126).

While AIMS also showed a good agreement with PAM50 (accuracy = 73.5%, kappa = 0.616: 95% CI = 0.595–0.637), AIMS assigned more patients to the Normal-like subtype (18.0%) compared to PAM50 (6.6%). Patterns of survival were similar to those observed for MiniABS (Figure 3C).

The only method that enables PAM50 subtyping of standalone single sample data is using the “none” option in the genefu package that processes the data without standardization. PAM50none showed a moderate agreement with the author-provided PAM50 (accuracy = 75.9%, kappa = 0.599: 95% CI = 0.574–0.619). Agreement between MiniABS and PAM50none was good (accuracy = 88.8%, kappa = 0.768: 95% CI = 0.747–0.789) compared to only moderate agreement between AIMS and PAM50none (accuracy = 69.4%, kappa = 0.506: 95% CI = 0.484–0.529). Again, the survival pattern for PAM50none was similar to those observed for MiniABS (Figure 3D).

Due to the use of a small number of genes, MiniABS could be tested on all datasets from the 10 studies and showed an average accuracy of 86.54% without Normal-like (Figure 4A,B), which was ~8.82% higher than that for AIMS (77.72%).

Of particular note, AIMS could not be applied to the five low-throughput datasets due to its requirement for a large number of genes (NanoString and qRT-PCR, marked “N/A” in Figure 4A,B). The average accuracy of MiniABS in the common datasets without Normal-like was 85.15%, outperforming AIMS (77.72%) for both the RNA-seq and microarray platforms (85.47% vs. 83.41% in RNA-seq, and 84.68% vs. 69.20% in microarray) when regarding PAM50 as the gold standard (Figure 4B). The high average accuracy for NanoString (91.11% and 84.98% without/with Normal-like, respectively) was unexpected and noteworthy, because no such datasets were included in the training phase.

## 3. Discussion

In order to develop a true absolute SSC for breast cancer, we used a stepwise approach of selecting differentially expressed genes among subtypes and used ratios of those genes as an input for machine learning with PAM50 subtype assignment as a gold standard reference in the TCGA breast cancer dataset. The resulting 11 genes of MiniABS showed a robust performance regardless of the technology platform to measure gene expression. MiniABS may have utility when comparing small sample size clinical trial cohorts with biased subtype enrichment (for example, ER+ tumors only) due to its independence from cohort composition. Parker’s method is challenging because it is difficult to construct an ER balanced dataset in the absence of IHC data or when imbalance is severe in the subtype composition of the cohort. In addition, in the case of actual real cohort data, application of Parker’s method may be limited depending on the purpose and circumstances of the study (e.g., drug responsive studies focused on a cohort of only ER+/HER2− cases or only HER2+ cases). Furthermore, the process of centering genes can also sensitively affect the results. While AIMS is also an absolute SSC, it requires more genes and therefore, may be difficult to apply to low throughput technology platforms such as qRT-PCR.

Selecting a gene subset for classification with a brute-force manner requires an extremely high load of computation (Appendix B, Appendix A), which often leads to false discovery of random gene sets. In this study, three major heuristic approaches were adopted to alleviate this problem. First, we used a stepwise reduction when minimizing the gene set. Instead of enumerating all possible cases, we first tried to find the upper-bound of the gene number (13 in MiniABS) at which the model performance converges, and then, searched for the minimal gene set. Second, limiting the pool of total genes based on their functional association further reduced the search space and the risk for selecting a false gene. By using the PAM50 genes, we greatly reduced the number of cases by a factor of ~10^20^, without a loss in accuracy. Lastly, pre-selection of four seed genes was done based on the same rationale (i.e., securing functional relevance and model generality while reducing search space). Again, there was no loss of model performance caused by this step. Overall, we were able to reduce the number of cases to ~10^9^ from ~10^6020^ (Appendix B), which is an easily addressable size, with the aid of ssDEG analysis, and we were able to avoid local optima problems. Therefore, by using the PAM50 gene, which is well-studied in the breast cancer subtype, and four markers used in clinical practice, rather than any gene, the intention was to reflect the mechanism of the breast cancer subtype more, and will not be of limited temporal performance seen only in the training dataset. MiniABS pursued an effort to improve performance by reflecting the mechanism of the breast cancer subtype, while losing less information on the input used in the model than AIMS using the binary rule of gene pairs without considering the biological significance.

The genes used in MiniABS are known to be involved in biologic characteristics, such as proliferation (MYBL2, MKI67, and CEP55), HER2 signaling (ERBB2), growth factor signaling (FGFR4), ER signaling (FOXA1, MLPH, ESR1, and PGR), and Basal phenotype (KRT17, and SFRP1) [19,20,21]. Additionally, SFRP1 is a known Basal-like marker, and MYBL2 is a LumA-specific cellular proliferation marker [8,22]. From this study, based on the distribution of the top PGER values across the subtypes, we observed that each subtype was able to be characterized better with subtype-specific different gene expression ratios than single gene-specific expression (Appendix A and Appendix A, top *p* values within 10^−58^–10^−59^, compared to 10^−19^–10^−50^ for a single gene). For example, Her2E and LumA subtypes could be partially characterized by ESR1/ERBB2 and PGR/MYBL2 (Appendix A) and LumA and LumB with MYBL2/FOXA1, SFRP1/MYBL2, and MYBL2/MLPH (Appendix A). Therefore, studying differential gene expression ratios and inter-gene interaction among breast cancer subtypes may help us to further understand them.

Currently, the only two regulatory approved subtyping tests for breast cancer are Prosigna (based on the nCounter platform) and BluePrint (based on microarray or RNA-seq). These two subtypers use proprietary algorithms and do not completely agree with research-based PAM50 assay results. More importantly, the agreement between the two was only moderate (kappa = 0.55: 95% CI = 0.45–0.64, *N* = 302) in the OPTIMA Prelim trial that prospectively compared multiple prognostic tests for breast cancer [17].

While MiniABS subtype assignment does not completely agree with PAM50, which is regarded as the gold standard, and assigns more patients to the LumA subtype (especially LumB to LumA), intriguingly, there was no survival difference between MiniABS LumA and PAM50 LumA patients treated with endocrine therapy in a large validation cohort. PAM50 LumB patients misclassified as LumA by MiniABS showed intermediate survival outcomes between LumB and those classified as LumA by both PAM50 and MiniABS. This was true for AIMS and PAM50none. Thus, the discrepant cases may represent tumors with intermediate characteristics between LumA and LumB and do not reflect true misclassification.

## 4. Materials and Methods

### 4.1. Method Overview

The overall workflow is shown in Figure 1. Machine learning classifiers were trained with un-normalized mRNA expression data (without normalization among different samples) from TCGA annotated with PAM50 subtypes. To select informative genes for classification, genes with a subtype-specific expression pattern were extracted. The expression ratios between all possible informative gene-pairs (PGER) were calculated and were used as inputs of machine learning classifiers. In the training phase, the classifier with the smallest gene set, but maximum classification power, was selected as an initial model, which was further tested on an independent validation cohort to confirm its accuracy. When genes were ranked based on PGER (feature) importance (Gini score index) from the initial model in the training phase, we noted a remarkable decrease in importance between the top 7 and 8 genes. Thus, model optimization to reduce the size of the gene set was performed by sequentially deducting genes one by one from the lowest rank (13th) to the 7th gene, after which the model was rebuilt without the removed genes in the same way applied in the training phase using the PGER matrices of these genes, and the best performing model was selected as the final model for MiniABS. The TCGA test set was then used to test the accuracy of the finalized MiniABS before moving on to the next step of independent validation across different technology platforms. We further tested it on an independent validation cohort to confirm its accuracy. MiniABS is publicly available at https://sourceforge.net/projects/miniabs/.

### 4.2. Training Dataset

For a discovery set, we used un-normalized RNA-seq and microarray-based gene expression data from 432 breast cancer samples from The Cancer Genome Atlas (TCGA BRCA) [7]. For model construction, the expression values were log2-transformed. According to TCGA annotation with PAM50, the 432 samples consisted of 76 Basal-like, 50 Her2E, 194 LumA, 105 LumB, and 7 Normal-like subtypes. Initial training was performed using RNA-seq and microarray data were used to examine the cross platform applicability of the developed algorithm. For RNA-seq data, we downloaded TCGA RNAseqV1 level 3 data with RPKM expression units. For microarray data, we downloaded the Agilent 224K Gene Expression Microarray Level 1 data (Agilent two-channel using UNC custom Microarrays). Agilent arrays, like the processing of AIMS [4], used only the channel of the tumor samples, and then, subtracted background intensities from this value. In the process of selecting a probe representing a gene, we selected the probe with the maximum expression value from each sample, rather than using the standard deviation of probe expression levels to avoid the effect derived from studied cohort, such as the number of samples and composition of subtypes, as Paquet et al. described previously [4].

### 4.3. Feature Selection and Input Data (PGER Matrix) Preparation

To identify informative genes for the classification of subtypes, a ranked list of genes significantly up- or down-regulated in only one of the five subtypes (subtype-specific differentially expressed genes, ssDEGs) was obtained by performing a Wilcoxon rank-sum test, with a Benjamini–Hochberg false discovery rate-corrected *p* value of less than 0.005.

Since classification modeling based on pairwise comparisons of this many ssDEGs would demand too much computing power, we employed a pragmatic approach to reduce the initial search space and the number of trials in order to build a robust and possibility interpretable classifier (Appendix B). Thus, four well-known subtype marker genes, ESR1, PGR, ERBB2, and MKI67 [8,9], were used as seed genes. To the list of four seed genes, five genes from each rank were added stepwise from the 1st to the 5th level. In other words, we defined the kth-level subset as a set of top-k ranked ssDEGs genes for each of the five subtypes: for example, the first-level subset was {MLPH, FGFR4, CEP55, KRT17, ERBB2}, and the second-level was the first-level subset + {FOXA1, GRB7, MYBL2, SFRP1, KRT14} (Figure 1). In total, five subsets were prepared by increasing k from 1 to 5 and were defined as the input gene sets (seed and top-n ranked ssDEGs).

To generate the pairwise gene expression ratio (PGER) matrix to be used as an input for machine learning training, for each input gene set, log_2_(RPKM+1)-transformed expression ratios between all possible combinations of gene pairs were calculated. Given two raw expression (e.g., RPKM in RNA-seq) values *e_i_* and *e_j_* of genes *g_i_* and *g_j_*, the pairwise gene expression ratio (PGER) *r_ij_* is calculated by:$$rij=\log2\left(\frac{ei+1}{ej+1}\right),\;1\leq i<j\leq n$$
where *n* is the total number of genes in the input gene set. We assumed that the gene expression difference at this level cannot be differentiated from experimental noise; therefore, we further transformed *r_ij_* by introducing a slack margin variable α, wherein any *r_ij_* whose absolute value was smaller than α is counted as zero.
0, if |*r_ij_*| ≤ α
*r’_ij_* = *r_ij_* − α, if *r_ij_* > α
*r_ij_* + α, if *r_ij_* < −α

Finally, an m × *n*(*n* − 1)/2 matrix of gene expression ratios (*r’_ij_*) was prepared for input, for each ssDEG subset level (1 to 5) and α (0.00, 0.01, 0.05, 0.10, 0.15, 0.20, and 1.00), wherein m is the number of samples in the training set and (_n_C_2_) = *n*(*n* − 1)/2 is the number of possible gene pairs.

### 4.4. Training and Optimization of the Classifier

In the training phase, TCGA BRCA data were split into training and test datasets (4:1) using a stratified random sampling method in Caret [23] package. Model training with five-fold cross-validation repeated 100 times was performed on the training dataset using Caret [23] in the R package. Training was attempted using four different machine learning algorithms: support vector machine (SVM) with radial kernel, classification and regression tree (CART), random forest (RF), and naïve Bayes (NB). For each model, a five-fold cross-validation, along with hyperparameter tuning using the “tuneLength” option, was performed repeatedly 100 times. Through this process, the optimal combination of adjustable hyperparameters in each model was automatically tuned to select the model with the best performance. In the training phase, the classifier with the smallest gene set, but maximum classification power, was selected as an initial model.

The importance of each gene was inferred and ranked using Gini index scores, which reflect the importance of contributing features to the model, assigned to the features (PGER) of the initial model determined in the training phase. When genes were ranked based on importance, we noted a remarkable decrease in importance between the top 7 and 8 genes. Thus, model optimization to reduce the size of the gene set was performed by sequentially deducting genes one by one from the lowest rank (13th) to the 7th gene, after which the model was rebuilt without the removed genes in the same way applied in the training phase using the PGER matrices of these genes. The best performing model was selected as the final model for MiniABS. The TCGA test set was then used to test the accuracy of the finalized MiniABS before moving on to the next step of independent validation across different technology platforms.

### 4.5. Independent Validation and Processing

The validation sets consisted of 5816 samples acquired from 10 independent studies. Detailed dataset information and acquisition processes are provided in Appendix A [8,18,24,25,26,27,28,29,30,31]. The validation datasets were downloaded from the NCBI’s Gene Expression Omnibus (GEO) [32] and original publications. For Affymetrix array, in order to use the absolute expression values in a given sample, raw data (CEL files) were processed using Robust Multi-array Average (RMA) normalization per single sample with the R/Bioconductor affy [33] package. We selected the probe per gene with maximum expression values from each sample. For the RNA-seq data, GSE96058 and GSE81538, the log2(FPKM+0.1) values provided by the authors were used. As the input expression data for array data for AIMS application, KRT17 and CENPU genes were used as 205157_s_at, 212236_x_at, and 218883_s_at, respectively, and then, all genes were converted into Entrez IDs as required for AIMS. The PAM50 intrinsic subtype information was obtained from the files provided in the original papers of the authors, except for GSE41998. For GSE41998 data, we used the PAM50 subtype provided by Prat et al. [34]. The authors of the original paper used Parker’s research-based PAM50 to assign the molecular subtype, except for two cohorts, GSE60788 (*N* = 22) and Priedigkeit et al. [31] (NonoString, *N* = 40), consisting of a small sample of 10 validation datasets. The genefu [35] R/Bioconductor package with the default and “none” parameters was used to identify the subtypes of samples from RNA-seq datasets.

### 4.6. Performance Assessment with and without Normal-Like

We calculated the accuracy of MiniABS both with and without the Normal-like subtype for the validation dataset, because the Normal-like subtype is considered a controversial subtype in breast cancer studies, which have failed to determine whether to define it as a genuine breast cancer subtype: Normal-like subtype has been suggested as tumors containing a large amount of contamination from normal tissue [36,37]. When applying MiniABS to the validation dataset, we used all models, consisting of seven α values, and then, determined the most predicted subtypes as the final subtype. In rare cases, there were instances in which two subtypes were assigned because the probabilities for the two subtypes were the same. Even if the correct PAM50 subtype was assigned, we considered this as misalignment and calculated accuracy accordingly.

### 4.7. Modeling with Seed Genes, ssDEGs, and Intrinsic Gene Set

To evaluate the effect of using the PAM50 gene set, the models were also constructed and evaluated using an “intrinsic gene set” and ssDEGs from all genes in the RNA-seq dataset, instead of only the PAM50 gene set. The intrinsic gene set used by Parker et al. was downloaded from GeneSigDB [38]. To evaluate the effect of using the seed genes, the models were also assessed without the seed genes. The set of genes used in model construction and accuracy results can be seen in Appendix A.

### 4.8. Feature Analysis

The Kruskal–Wallis test was performed to determine whether each feature, the pairwise gene expression ratios, was significantly different among the subtypes. By using the median value of each gene for each subtype, the pattern of expression of genes among the subtypes used in the model was confirmed. All analyses were carried by using R statistical software, version 3.2.5.

### 4.9. Statistical Analysis

The accuracy and Cohen’s K were calculated to compare the classifiers (PAM50 provided by the authors, PAM50none using the “none” parameter of genefu, and AIMS). Kaplan–Meier and Cox regression survival analyses were performed with overall survival as the end point. All calculations were performed with R, version 3.2.5.

## 5. Conclusions

We developed a single breast cancer sample subtype classifier (MiniABS) that could accurately be subtyped from single patient-derived gene expression data of only 11 genes with better efficiency and without platform and normalization bias. The performance of it surpassed previously developed classifiers and can be applied on multiplatform data. We anticipate that MiniABS could be developed into a practical test and accelerate translational research.

## Figures and Tables

**Figure 1 cancers-12-03506-f001:**
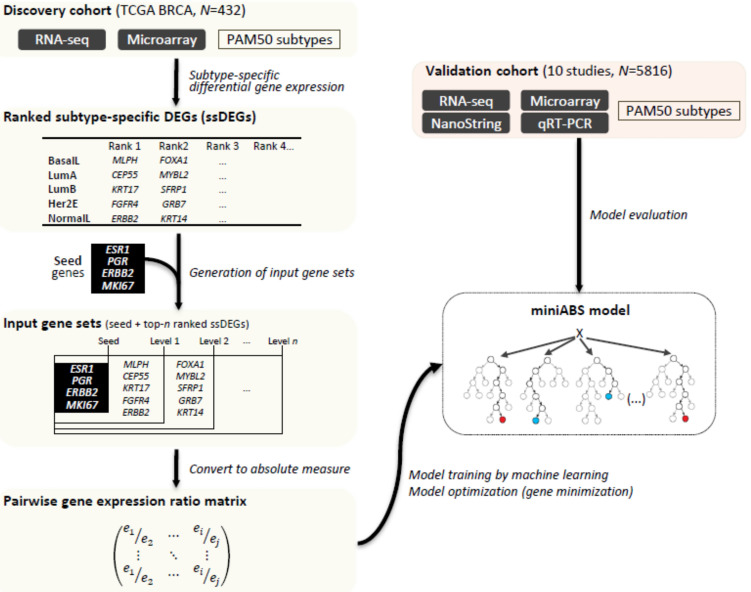
Overview of MiniABS. To identify the features used to classify the molecular subtypes of the model, subtype-specific DEGs (ssDEGs) were identified using the Wilcoxon rank-sum test. We set up an input gene set for gene selection with the ability to classify subtypes with a minimal gene set. Four well-known machine learning models with pairwise gene expression ratios (PGERs) were used to learn and optimize the models. The model with the best accuracy was selected. The performance thereof was evaluated by applying the model to the test set from the discovery set. Finally, we evaluated whether the final selected model (MiniABS) works well with an independent validation dataset and has robustness.

**Figure 2 cancers-12-03506-f002:**
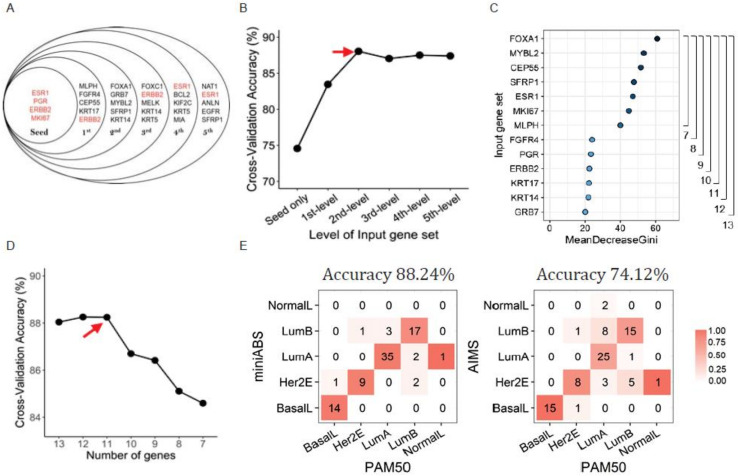
Optimization and performance of MiniABS. (**A**) Input gene set. The red lettering corresponds to the seed genes. (**B**) Cross-validation accuracy according to levels in the input gene set. The red arrow corresponds to best accuracy. (**C**) Mean decrease in Gini for the input gene set. Degrees of decrease in purity when splitting occurs during training after the top seven genes are reduced. (**D**) Cross-validation accuracy according to number of genes. The red arrow corresponds to best accuracy. In the case of modeling with fewer than 11 genes, we can see that the accuracy decreases sharply. (**E**) Confusion matrix of the test set (*N* = 85) of TCGA BRCA. The numbers in the boxes indicate the number of samples, and the units on the color bar represent the concordance rate per subtype. The MiniABS clearly shows better overall accuracy than AIMS, and in particular, the LumA was correctly assigned the most with MiniABS than AIMS.

**Figure 3 cancers-12-03506-f003:**
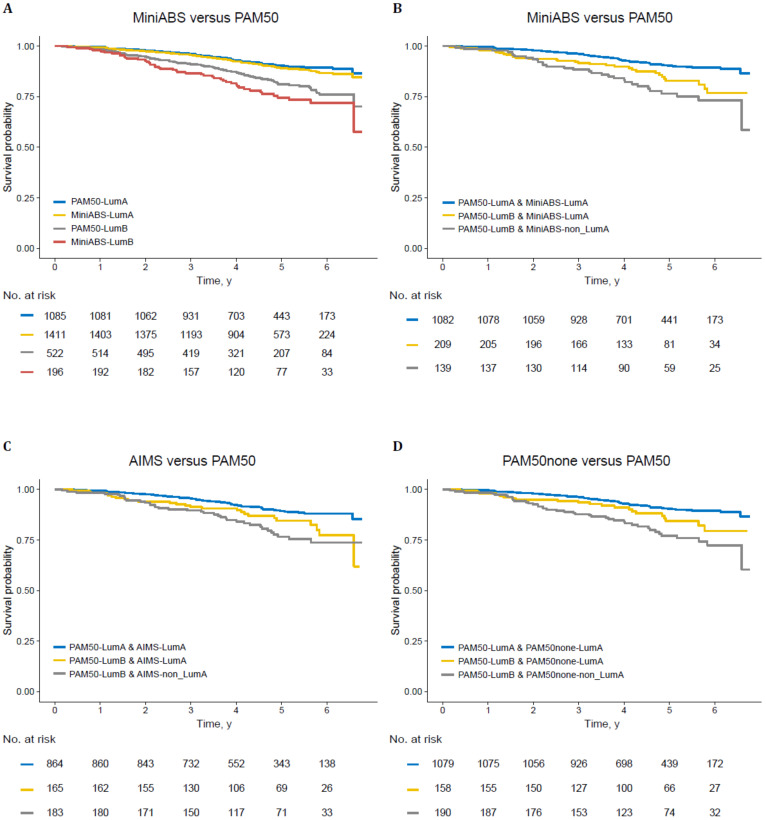
Kaplan–Meier survival analysis of patients treated by endocrine therapy. (**A**) Survival plot for LumA and non-LumA of MiniABS versus PAM50. Survival plot for PAM50 and (**B**) MiniABS; (**C**) AIMS; (**D**) PAM50none. PAM50none is a subtype of the sample identified by the genefu package using the “none” option.

**Figure 4 cancers-12-03506-f004:**
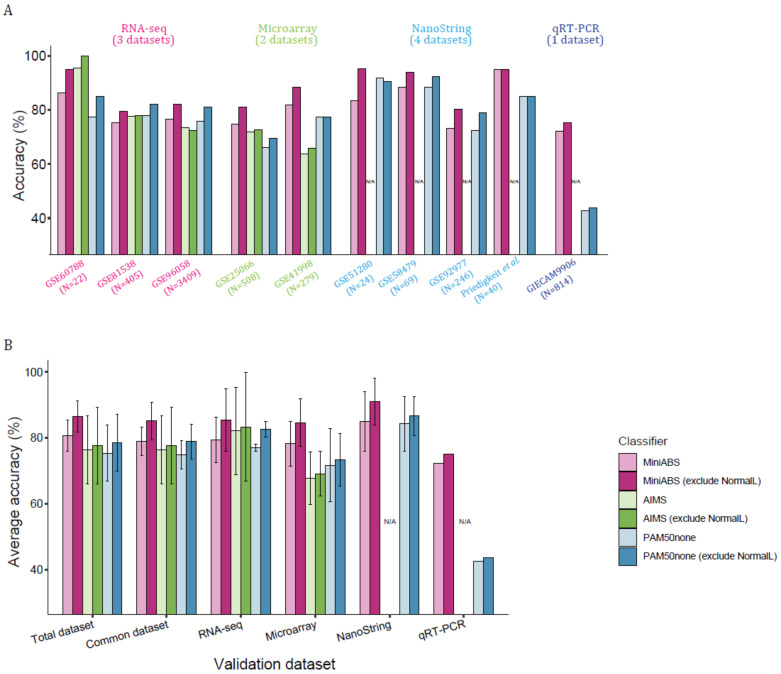
Accuracy in the validation dataset. (**A**) Accuracy in the validation dataset. The numbers in parentheses beneath the datasets correspond to the number of samples. The lack of genes in NanoString and qRT-PCR data prevented the application of AIMS. Bars indicate the accuracy of MiniABS (pink bars), AIMS (green bars), and PAM50none (blue bars). The bars filled with light colors denote the accuracy for the five subtypes, including the Normal-like subtype, and the dark color bars correspond to the accuracy calculated after removing the Normal-like subtype. (**B**) Average accuracy in the validation dataset for the total dataset, common dataset (datasets that could apply to both MiniABS and AIMS classifiers), and each individual platform. Error bars indicate 95% confidence intervals (95% CI).

**Table 1 cancers-12-03506-t001:** Top ssDEGs for five subtypes.

Rank	Basal-Like ssDEGs			Her2E ssDEGs			LumA ssDEGs			LumB ssDEGs			Normal-Like ssDEGs		
	Symbol	*p*-Value	FDR	Symbol	*p*-Value	FDR	Symbol	*p*-Value	FDR	Symbol	*p*-Value	FDR	Symbol	*p*-Value	FDR
1	MLPH	4.0 × 10^−41^	8.2 × 10^−37^	FGFR4	4.6 × 10^−19^	4.3 × 10^−15^	CEP55	2.4 × 10^−50^	1.2 × 10^−46^	KRT17	2.0 × 10^−20^	6.8 × 10^−17^	ERBB2	5.4 × 10^−3^	2.4 × 10^−1^
2	FOXA1	9.6 × 10^−40^	6.6 × 10^−36^	GRB7	5.6 × 10^−16^	7.7 × 10^−13^	MYBL2	7.0 × 10^−49^	2.4 × 10^−45^	SFRP1	7.7 × 10^−20^	1.6 × 10^−16^	KRT14	2.3 × 10^−3^	2.4 × 10^−1^
3	FOXC1	1.0 × 10^−38^	3.4 × 10^−35^	ERBB2	1.4 × 10^−15^	1.6 × 10^−12^	MELK	3.8 × 10^−47^	5.2 × 10^−44^	KRT14	5.7 × 10^−18^	4.3 × 10^−15^	KRT5	3.0 × 10^−3^	2.4 × 10^−1^
4	ESR1	1.0 × 10^−35^	6.5 × 10^−33^	BCL2	5.2 × 10^−14^	2.8 × 10^−11^	KIF2C	7.5 × 10^−47^	9.6 × 10^−44^	KRT5	1.6 × 10^−17^	8.7 × 10^−15^	MIA	3.2 × 10^−3^	2.4 × 10^−1^
5	NAT1	1.4 × 10^−35^	8.1 × 10^−33^	ESR1	3.6 × 10^−12^	9.5 × 10^−10^	ANLN	1.2 × 10^−46^	1.5 × 10^−43^	EGFR	1.1 × 10^−16^	5.0 × 10^−14^	SFRP1	4.9 × 10^−3^	2.5 × 10^−1^

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
