# Peer review of "An Improved, Assay Platform Agnostic, Absolute Single Sample Breast Cancer Subtype Classifier"

_cancers, 2020, doi:10.3390/cancers12123506_

Round 1

Reviewer 1 Report

An Improved, Assay Platform Agnostic, Absolute Single Sample Breast Cancer Subtype Classifier

In this paper, the authors sought to develop a platform-independent breast cancer subtype classifier based on the minimal number of genes using machine learning. In doing so, the authors developed and validated a random forest single sample classifier (MiniABS) using just 11 genes in silico. The paper was well thought out and well written. However, for a publication in Cancers, it would be nice to see validation of the 11 genes in a low-throughput assay by qRT-PCR of blindly selected cases with different cancer subtypes. The ultimate usefulness of MiniABS is its translational application, particularly that the bioinformatics and machine learning techniques, while interesting and powerful, are not new.

Other minor comments/suggestions to improve the quality of the paper:

Line 42: “closet correlation” should be revised to “closest correlation”.

Line 45: Please revise “…relativistic nature; 1) Due to the need…” Consider split the sentence into two, at the “;”.

Line 136: Table 1 legend indicates red and blue colors for over- and down-regulated genes, respectively, but the colors are not visible on the table.

Line 154: Please revise “...overall than AIMS, but also LumA.” Unclear what was meant.

Line 199: Please delete one “only” from …”and only assigned only 0.1%...”

Line 362: Please remove “=” from the …”(=0.00, 0.01, …)”

Line 380: Please split the sentence into two at “:”, so it becomes …’inputs. In short, …”

Reviewer 2 Report

In this manuscript the authors proposed a method for better prediction of breast cancer subtypes based on gene information. The novelty for this method is that, instead of using absolute or relative expression amount, they employed pairwise ratios (PGER) for prediction purpose. They also select about 11 genes that can achieve nearly 90% prediction accuracy in cross-validation dataset and 85% in independent validation datasets. I am a bit surprised that such small amount of genes can achieve such a good performances and think that this work could be useful for profiling cancer subtypes from gene expression data. But there are still a few things that I hope the authors could clarify. Below please find my reviews.

  1. Pairwise gene expression ration (PGER) is perhaps the most important concept in this paper since the application of PGER instead of normalized expression amounts provides better results. However I still have problems in imagining how the data would look like. In normal classification tasks, every data entry has a value in every feature. For example, in the iris dataset all iris flowers have petal width, petal length, sepal width, and sepal length values. If I were to transform this dataset into PGER, however, I need to calculate all possible combinations of rij, thus generating as many as six (or nine; questioned below) features. The authors further stated in the method that “PGER matrices (input gene sets [N=5] x alpha values [N=7])”, which confused me regarding to what the classification matrix really looks like. I hope the authors can spend some effort into explaining this concept and the actual data outlook in a much more clear way.

  1. I don’t know why the the number of possible gene pairs is n(n+1)/2. Unless ei=ej, otherwise, according to the PGER formula (log2(ei/ej+1)), log2(ei/ej+1) != log(ej/ei+1). Not sure if I misunderstand anything but I hope that the authors could further clarify this part.

  1. Is it log2(ei/(ej+1)) or log2((ei/ej)+1)? And why +1?

  1. It’s really confusing to see that the authors include all matrices generating using different alpha into training--technically alpha is pre-determined and used to generate one and only one matrix. Please either clarify if this is not the case, or explain why.

  1. Perhaps due to the way of writing, I feel that the methodology part is not completely in order. For example, in section 4.4, the authors first introduce how training and test datasets was split and how cross-validation was conducted; they then describe how PGER matrix was generated and fed into the algorithms. This creates an impression that the training/testing of the models is done BEFORE generating PGER matrix. I can probably guess this is not the case though. Please go through the article and check whether such scenarios occur anywhere.

  1. Only 55 PGERs generated from 11 genes instead of n(n+1)/2?

  1. More citations are needed for the introduction. For example line60-62 and line 64-67 warrants citations. And this are not the only places lack citations. Please carefully go through the manuscript and cite sources properly.

  1. (line 380) one of the “machine learning” term is redundant.

  1. The first appearance of the PGER term (line 112) is not explained.

  1. Maybe a few discussion of the selected genes in the Discussion can be more helpful in understanding what are the features (say, the 11 genes) and how are they related to breast cancer subtypes.

Round 2

Reviewer 2 Report

The authors have addressed most of my questions and concerns. I have only one minor comment regarding to the number of possible PGERs. I understand the author's explanation that PGER is like gene expression, in one is trying to look for degree of relationships between two genes. This is fine. However according to the formula (line 363 in the revised manuscript) every pair between i and j will be calculated given i!=j. In other words, this formula is going to create nine PGERs given four features instead of six. Please revise this formula so that correct number of pairs can be obtained (or maybe a simple algorithm, etc.).
